# Prevalence, patterns and determinants of peripheral neuropathy among leprosy patients in Northeast Ethiopia: A retrospective study

**Endris Seid Muhaba**[1]*, **Soressa Abebe Geneti**[2], **Dereje Melka**[3], **Seid Mohammed Abdu**[1]

**1** Department of Biomedical Sciences, College of Medical and Health Sciences, Wollo University, Dessie, Ethiopia, **2** Department of Anatomy, School of Medicine, College of Health Sciences, Addis Ababa University, Addis Ababa, Ethiopia, **3** Department of Neurology, School of Medicine, College of Health Sciences, Addis Ababa University, Addis Ababa, Ethiopia

* idrisibnuseid@gmail.com

## Abstract

### Background

Leprosy, also known as Hansen's disease, is a bacterial disease caused by *Mycobacterium leprae* (*M. leprae*). Leprosy is believed to have originated initially in Eastern African regions and spread worldwide through the migration of people. Leprosy causes skin and nerve infections. It has been investigated that the Hansen's bacillus, *M. leprae*, predominantly targets peripheral nerves.

### Methodology/Principal findings

The present study reviewed charts of 380 patients with complete records fulfilling the inclusion criteria. An institution-based retrospective cross-sectional study design was employed. The study was conducted at Boru meda General Hospital, located in the South Wollo zone, Amhara region, Ethiopia from September 2019 to August 2021. In the current study, the prevalence of peripheral neuropathy among leprosy patients is found to be 60%. Male sex, advanced age, presence of leprosy reactions, presence of more than four skin lesions, longer duration of the disease, and MB leprosy were the risk factors associated with leprosy peripheral neuropathy. Sensory impairment was the most common presentation followed by motor impairment.

### Conclusions/Significance

In Ethiopia, data is scarce regarding the prevalence, pattern and determinants of leprosy peripheral neuropathy. Hence, this study was intended to assess the prevalence, pattern and determinants of leprosy peripheral neuropathy. The result of this study would be served as an important input to develop recommendations that inform some clues for future researchers in this area. This study showed high prevalence of leprosy peripheral neuropathy among registered leprosy patients reflecting how much significant the study is.

**Data availability statement:** All data are in the manuscript and/or supporting information files.

**Funding:** The author(s) received no specific funding for this work.

**Competing interests:** The authors have declared that no competing interests exist.

## Author summary

Leprosy is a chronic, infectious disease caused by *M. leprae*. It is characterized by loss of sensation, muscle weakness, Skin lesions, nerve damage, and deformities. This study investigates the prevalence, patterns, and determinants of peripheral neuropathy in individuals with leprosy. We found a notable prevalence of 60% for leprosy-associated peripheral neuropathy. The most common pattern observed was sensory neuropathy, indicating a significant impact on sensory functions in affected individuals. The study further explored various determinants contributing to this high prevalence, offering insights into potential areas for targeted intervention and management strategies.

## Introduction

Leprosy is a bacterial disease caused by *Mycobacterium leprae (M. leprae)*, which results in skin and nerve infections [1]. Leprosy is known for preferentially affecting the nerves of the extremities, as well as the mucous membrane lining of the nasal cavities, eyes, and upper respiratory tracts [2].

Until the late 1940s, the reason for paralysis in Hansen's disease patients remained unknown. Paul Brand (1914–2003), a prominent leprologist and hand surgeon, made a significant breakthrough by discovering that Hansen's bacillus, *M. leprae*, predominantly affects peripheral nerves [3].

Leprosy, one of the neglected tropical diseases, continues to be an important public health concern in several developing countries [4].

The World Health Organization (WHO) reported a high rate of new leprosy cases globally. Between 500,000 and 750,000 new cases detected annually. In 2005 alone, over 500,000 new cases were identified, equating to more than 1,400 cases daily or nearly 60 cases every hour [5].

Despite multidrug treatment (MDT) and WHO surveillance programs contributing to a fortunate 55% decrease in the global prevalence of leprosy, it remains a significant cause of peripheral neuropathy. In 2012, the prevalence of leprosy was highest in South east Asia(116 per 100,000), followed by Africa (53 per 100,000) and Central and South America(46 per 100,000) [6].

Ethiopia is the second most affected country in Sub-Saharan Africa (SSA), after the Democratic Republic of Congo. Annually, 4000–4500 new cases were diagnosed at health facilities between 2004 and 2010 [7]. Nevertheless Ethiopia achieved the WHO leprosy elimination target of 1 case per 10,000 population in 1999, the occurrence of new cases remains a challenge [8].

Ethiopia is one of the nations with the greatest number of newly diagnosed cases of grade II disability [9]. In 2013, 4374 (4028 MB and 346 PB) new cases of leprosy were reported. Of these, 466 (or 10.65%) were children, and 361 (8.25%) of the newly diagnosed cases had a grade II disability [10]. As discussed by a recent study, out of 57 new patients presented over the course of three months, nearly 60% had grade II disability of leprosy [11]. 3,970 new cases of leprosy were reported in Ethiopia in 2015;14.2% of these cases were children and 10.6% of new cases had a disability grade II at the time of diagnosis [12].

By the end of 2001, Ethiopia's general health care facilities have successfully integrated the leprosy control program, making sure that patients are diagnosed early and complete the full course of MDT successfully without concurrent disability [13]. But the passive case detection or self-reporting system of the integrated leprosy control program has led to an increase in

hidden and un-diagnosed leprosy cases within the community, thereby contributing to more severe deformities and disabilities [14]. Additionally, the absence of specialized expertise among general healthcare personnel has hindered access to diagnostic and treatment services [15].

In Ethiopian culture, there are a number of false traditional or religious beliefs and misconceptions around leprosy. A common misunderstanding is that leprosy is thought to be a curse from God or an ancestor, a punishment for crimes, or the product of witchcraft and can spread via touch or casual contact. Such misunderstandings intensify stigma and discrimination against those affected [16].

Peripheral nerve damage due to leprosy causes loss of sensation and tissue damage, leading to infectious disabilities. This can gradually result in self-amputation of the hands and feet. Blindness is also a potential complication [17]. Regarding disability, it is estimated that 3 million people are disabled by leprosy worldwide [18]. A systematic review and meta-analysis of 32 studies found that physical disabilities are strongly associated with male sex, multibacillary leprosy and leprosy reactions [19].

leprosy is curable, and physical disabilities that impact an individual's social and working life can be prevented with early treatment [20]. The typical late-stage clinical features of leprosy include sensory and motor loss in the face and limbs. It is these aspects of the disease that contribute to its well-known severe social consequences [21].

As estimated through clinical case series, about 4%–8% of all leprosy cases are limited to peripheral nerves, posing a diagnostic challenge [22]. Due to significant peripheral nerve involvement, leprosy has become a stigmatizing public health concern of considerable magnitude [23].

Globally, prior to diabetic neuropathy, leprosy used to be the most common peripheral nerve disorder. Despite affecting both the central nervous system(CNS) and peripheral nervous system(PNS), leprosy typically involves peripheral nerves [24]. Leprosy affects specific nerves such as the facial nerve, ulnar, median, and radial cutaneous nerves in the arm and hand, the posterior tibial and the sural nerves in the leg and foot as well as the lateral popliteal nerves [25].

In spite of being a chronic infectious disease of significant public health importance and one of the leading causes of permanent physical disability, the decline in prevalence following MDT has led to neglect of leprosy [26].

Despite imposing such a significant societal burden, studies on the prevalence, patterns, and risk factors associated with leprosy-related peripheral neuropathy are limited. Therefore, this study will offer valuable information about the prevalence, patterns, and associated factors of leprosy-related peripheral neuropathy for decision-makers, particularly in the study area.

## Methods

### Ethics statement

The study was conducted with the utmost respect for patient privacy and confidentiality. The data used in this study was taken from existing patient records, which were fully anonymized to ensure that no personally identifiable information was accessed or disclosed. Patient records were securely stored, with access limited to authorized personnel only, ensuring confidentiality and minimizing the risk of data breaches. Data collection procedures followed strict adherence to ethical standards. Ethical clearance was obtained from Institutional Review Board (IRB) of Addis Ababa University (AAU), College of Health Sciences, School of Medicine, Department of Anatomy. The ethical clearance and cooperation letter were also written to Boru meda General Hospital administration to get permission for data collection.

## Study area and study design

An institution-based retrospective study was conducted on patients diagnosed with leprosy and seen at the dermatology outpatient clinic of the Boru meda General Hospital, located in Dessie town, Amhara Regional State, Ethiopia. The hospital is one of the leprosy referral centers in Ethiopia. It was initially established by missionaries in 1954 mainly to provide care for leprosy and related complications. Gradually, the hospital started providing general medical services to the society. The hospital has 40 beds for leprosy and other dermatology cases. In addition, the hospital has three dermatology outpatient clinics. There are two dermatologists in the hospital: a tropical dermatology professional and a health officer with dermatology and leprosy training.

On average, approximately 20 leprosy patients are seen per month in the hospital. Two trained nurses with a BSc degree were assigned to collect the data from patients card room. The data collectors were supervised by two trained senior BSc degree nurses and the principal investigator (PI) during the process of data collection.

## Inclusion criteria

The study included charts of patients diagnosed with leprosy who attended Boru Meda General Hospital during the study period.

## Exclusion criteria

Patient charts that have a diagnosis of co morbid illness like DM and HIV, a documented history of chronic alcohol use and exposure to pesticides at workplace were excluded. Accordingly, 10 HIV and 25 DM, a total of 35 patients have been excluded.

## Study procedures

Outcomes definitions. Neuropathy (peripheral): functional impairment and/or structural damage to sensory, motor and autonomic nerve fibers within the peripheral nervous system detected by clinical nerve palpation (enlarged nerve, pain or feeling of electric shock); Pattern of leprosy neuropathy: a form of neuropathy in which leprosy affects the peripheral nerves. It can be sensory, motor or both [27]; Pauci-bacillary(PB): five or fewer lesions with no bacteria detected in the skin smear; Multi-bacillary(MB): more than five lesions or a bacterium is detected in skin smear, or both [28]; Neuritis: nerve damage with altered sensory and/or motor function with variable pain symptomatology; Acute neuritis: intense pain occurring spontaneously or during palpation of the nerve trunks, with evolution up to 3 months; Chronic neuritis: nerve damage of insidious onset and slow progression with variable pain symptomatology and over three months [29]; Silent neuritis: alteration of sensory and/or motor function in the absence of pain [30]; Neuropathic pain: presence of symptoms (pain, hyper-pathy, paresthesias) without progressive loss of neural, sensory and motor function [31]; Sensory damage: pain or paresthesias (numbness, tingling, needling, shock and burning) [32]; Motor impairment: Voluntary Muscle Test (VMT) result equals to or less than four [33]; Nerve Function Impairment (NFI): nerves that presented some alteration of sensory and/or motor function [34].

Patients and methods. The study included 380 medical records of leprosy patients who visited the dermatology clinic of the Boru meda General Hospital with a diagnosis of leprosy (both new and previously treated). Boru Meda hospital follows national guidelines for leprosy management. New cases were defined as patients who are presented with active disease and had never been treated for the disease. Previously treated cases are defined as patients

with a documented history of leprosy who visited the hospital for care of complications and disability.

Based on clinical and laboratory features, cases are diagnosed as pauci or multibacillary leprosy. pauci-bacillary applies for patients with one to five leprosy skin lesions and one nerve trunk enlargement whereas multi-bacillary (MB) is diagnosed when patients present with six or more skin lesions, less than six skin lesions which have a positive slit skin smear result and if there is involvement (enlargement) of more than one nerve. Pauci-bacillary cases are treated with two drugs for 6 months while multibacillary ones are treated with three drugs for 12 months [35]. The diagnosis of leprosy was made according to clinical criteria; the presence of ulceration, claw hand, foot drop and, voluntary sensory and motor tests.

A well-designed open data kit (ODK) form was used to collect the data. The Statistical Package for Social Sciences (SPSS), version 22.0, was used for analysis. Mean and SD were calculated for the continuous variables. Logistic regression analysis was employed to identify factors associated with leprosy peripheral neuropathy. First, bivariate logistic regression analysis was conducted for each independent variable. Variables with p-values of less than 0.2 were then considered as candidate variables for multivariate logistic regression to control possible confounders. Multivariate logistic regression where adjusted odds ratios (AORs) with their corresponding confidence intervals (CIs) were used to assess the strength of the associations between dependent and predictor variables at P-value ≤ 0.05 cut-off point. The model fitness was checked using the Hosmer-Lomeshow goodness fit test. Multicollinearity test was also checked at 10% Variance inflation factor (VIF).

## Results

### Socio-demographic and clinical characteristics

The study included a total sample of 380 leprosy patients. Of these, 203(53.4%) were males and 177 (46.6%) females. The mean age of leprosy patients was 41.25 ± 9.8. More than half, 234(61.6%) of patients were rural residents. One hundred nineteen (31.3%) had no formal education and smallest portion, 40(10.5%), were degree holders (Table 1).

A greater number, 263(69.2%) of patients suffered from leprosy reaction. Of which, 140(36.8%) experiencing type-1 reaction and 123(32.4%) type-2. The number of skin lesions ranged 0-7 and around one-third, 128(33.7%) had 2-3 skin lesions. The majority of patients, 154(40.5%) had 2-5 years duration with the disease. While 263(62.1%) of patients had PB leprosy, 144(37.9%) had MB leprosy. More than three quarters of patients, 291(76.6%) had no known contact history with leprosy inhabitants previously. Similarly, 285(75%) had no known family history of leprosy whereas 95(25%) had known familial history(Table 2).

Out of 228 affected individuals, 172(75.4) were presented with single nerve impairment, 56(24.6%) in multiple nerve impairments. Among the specific peripheral nerves, the most affected nerve was ulnar nerve, being encountered in 25(10.96%) followed by median nerve, revealed in 20(8.78%) of patients. Radial nerve was the least affected peripheral nerve 4(1.75%).

In our study, we observed leprosy reactions in relation to the type of leprosy. Overall, leprosy reactions were more frequent in Paucibacillary leprosy compared to Multibacillary leprosy. Type-1 reactions were more commonly seen in Paucibacillary leprosy, while type-2 reactions were more prevalent in Multibacillary leprosy (Table 3).

### Prevalence of leprosy peripheral neuropathy

In the current study, 228 leprosy patients had peripheral neuropathy with the overall prevalence of 60%. In terms of sex, leprosy peripheral neuropathy was recorded as 60.1% in males and 39.9% in females.

Table 1.  Sociodemographic characteristics of patients with leprosy at BGH 2022, Dessie, Ethiopia.

| Variables | Category | No. observations(%) |
|---|---|---|
| Sex | Male | 203(53.4) |
|  | Female | 177(46.6) |
| Age | < 30 years | 128(33.7) |
|  | 30-39 years | 82(21.6) |
|  | 40-49 years | 85(22.4) |
|  | ≥ 50 years | 85(22.4) |
| Occupation | Un employed | 97 (25.5) |
|  | Student | 42 (11.1) |
|  | Housewife | 46(12.1) |
|  | Farmer | 84 (22.1) |
|  | Self-employee | 78(20.5) |
|  | Government-employee | 33(8.7) |
| Marital status | Single | 105(27.6) |
|  | Married | 155 (40.8) |
|  | Divorced | 75 (19.7) |
|  | Widowed | 45(11.8) |
| Residency | Urban | 146(38.4) |
|  | Rural | 234(61.6) |
| Level of education | No formal education | 119(31.3) |
|  | Primary school | 74(19.5) |
|  | Secondary school | 73(19.2) |
|  | Diploma | 74(19.5) |
|  | Degree and above | 40(10.5) |

With respect to age, leprosy peripheral neuropathy was observed in 36.7% in <30 years, 58.5% in 30-39, 77.6% in 40-49 and 78.8% in ≥50 years.

## Patterns of leprosy peripheral neuropathy

The current study intended to present pattern of leprosy peripheral neuropathy in terms of sensory impairment, motor impairment and both sensory and motor impairments. The most common presentation was sensory impairment which was revealed in 24.5% of patients followed by motor impairment which was encountered in 15.8% of patients. As part of sensory impairment, neuropathic pain and nerve thickening affected 2.4% and 1.3% respectively. Both sensory and motor impairments were more common in males (Fig 1).

## Determinants of leprosy peripheral neuropathy

On bivariate investigation, some variables showed evidence of association with the outcome variable at a p-value of <0.2, hence included in the multivariate logistic regression analysis.

Variables which are expected to contribute in the development of leprosy peripheral neuropathy were entered into the binary logistic regression analysis to identify crude risk estimate. Accordingly, being male, age ≥ 30, being unemployed, farmer and self-employee, having rural residency, having no formal education, primary education and secondary education, having leprosy reaction, having ≥ 2 skin lesions and deep skin lesion, gradual mode of onset, staying with the disease for ≥ 2 years, having MB leprosy and having known history of contact with

Table 2. Clinical characteristics of patients with leprosy at BGH 2022, Dessie, Ethiopia.

| Variables | Category | No.observations (%) |
|---|---|---|
| Leprosy reaction | No | 117(30.8) |
|  | Type-1 | 140(36.8) |
|  | Type-2(ENL) | 123(32.4) |
| No of skin lesion/s | 0-1 | 115(30.3) |
|  | 2-3 | 128(33.7) |
|  | 4-5 | 54(14.2) |
|  | 6-7 | 83(21.8) |
| Level of skin lesion | Superficial | 240(67.2) |
|  | Deep | 117(32.8) |
| Nerve involvement | Single nerve | 172(75.4) |
|  | Multiple nerves | 56(24.6) |
| Mode of symptom onset | Sudden | 146(40.9) |
|  | Gradual | 211(59.1) |
| Duration of the disease (in years) | <1 | 94(24.7) |
|  | 2-5 | 154(40.5) |
|  | 6-10 | 107(28.2) |
|  | >10 | 25(6.6) |
| WHO classification of leprosy | PB | 236(62.1) |
|  | MB | 144(37.9) |
| Known history of contact with any patient affected by leprosy | No | 291(76.6) |
|  | Yes | 89(23.4) |
| Family history | No | 285(75.0) |
|  | Yes | 95(25.0) |

Table 3. Leprosy reaction vis-a-vis type of leprosy (PB, MB).

|  |  | WHO classification of the leprosy | | |
|---|---|---|---|---|
|  |  | PB | MB | Total |
|  | No | 89 | 28 | 117 |
| Leprosy reaction | Type-1 | 95 | 45 | 140 |
|  | Type-2 | 18 | 105 | 123 |
| Total |  | 236 | 144 | 380 |

leprosy patients revealed statistically significant association with leprosy peripheral neuropathy in the crude odds ratio analysis and selected as candidate for multivariate analysis (Table 4).

In the multivariate analysis, age, sex, duration of the disease, history of leprosy, leprosy reaction, number of skin lesion/s and WHO classification of leprosy appeared to be associated with leprosy peripheral neuropathy at a p-value of <0.05. Participants with age 40-49 years were 4.2 times more likely to develop leprosy peripheral neuropathy compared to patients younger than 30 years (AOR= 4.214;95% CI: 1.973-8.999).

Furthermore, participants with age 50 years and above were 5.3 times more likely to develop leprosy peripheral neuropathy as compared to patients whose age was less than 30 years (AOR = 5.287; 95%CI: 5.287 (2.473-11.301). The other associated factor was sex. Male participants were 2.2 times more likely to develop leprosy peripheral neuropathy as compared with females (AOR = 2.218; 95% CI: 1.230-3.998).

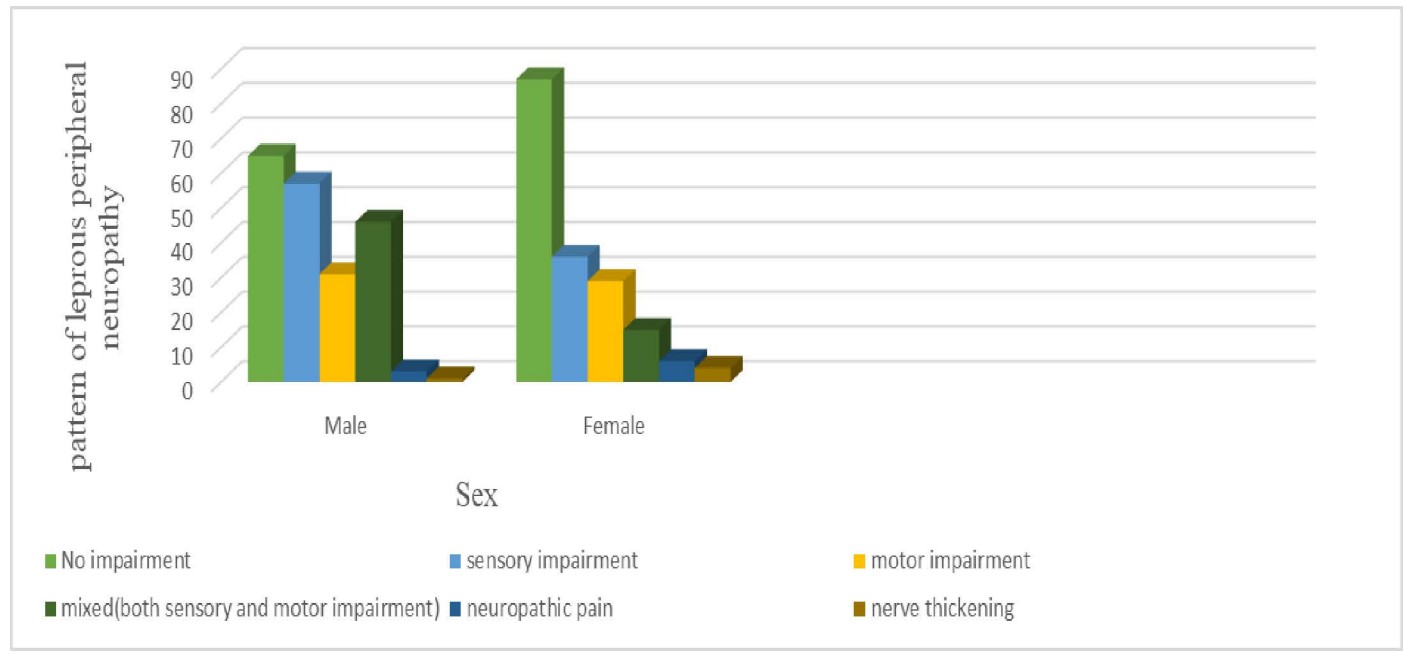

**Fig 1. bar graph of patterns of leprosy peripheral neuropathy in terms of sex.**

Those participants of greater than 10 years of duration with disease were 4.87 times more likely to develop leprosy peripheral neuropathy as compared with those with shorter leprosy history less than 1 year(AOR = 4.866; 95% CI: 1.130-20.958).

Similarly, leprosy patients who had type-1 and type-2 leprosy reactions were 1.9 and 2.5 times more likely to develop leprosy peripheral neuropathy when compared with patients with no leprosy reaction, (AOR = 1.982; 95% CI: 1.050- 3.742) and (AOR=2.509; 95% CI: 1.269-4.962) respectively.

Likewise, patients who had 4-5 skin lesions were 3.2 times and those who had 6-7 skin lesions were 3.6 times more likely to develop leprosy peripheral neuropathy as compared with those who had less than or equals to one skin lesion (AOR = 3.187; 95% CI:1.115- 9.108) and AOR =3.642; 95% CI: 1.411- 9.403) respectively.

Finally, patients who had MB leprosy were 3.6 times more likely to develop leprosy peripheral neuropathy as compared to those who were PB (AOR = 3.636; 95% CI:1.431-9.243) (Table 5).

## Discussion

The current study aimed to determine the prevalence and pattern of peripheral neuropathy and its correlates among patients with leprosy. According to the study findings, the prevalence of leprosy peripheral neuropathy was 60%. This prevalence is nearly in line with prevalences reported from studies conducted in Toronto and India,which were 55% and 63%, respectively [36,37].

A study from Carville, USA, reported a neuropathy prevalence of 67% [38]. This slight difference may be due to differences in the study population, diagnostic criteria employed, and methods of participant selection. The prevalence of leprosy peripheral neuropathy was slightly lower than the prevalence reported from the United Kingdom, which was 68% [39]. This difference could be explained by variations in study design, sampling methods, and tools used.

**Table 4. Bivariate analysis - Predictors of PN among leprosy patients at BGH 2022, Dessie, Ethiopia.**

| Variables | Category | Occurrence of LPN | | Bivariate analysis | |
|---|---|---|---|---|---|
| | | Yes | No | COR (95% CI) | p-value |
| **Sex** | Male | 137 | 66 | 1.962 (1.294-2.974) | .002* |
| | Female | 91 | 86 | 1 | |
| **Age (in years)** | < 30 | 47 | 81 | 1 | |
| | 30-39 | 48 | 34 | 2.433 (1.379-4.292) | .002* |
| | 40-49 | 66 | 19 | 5.987 (3.207-11.175) | .000* |
| | ≥ 50 | 67 | 18 | 6.415 (3.408-12.073) | .000* |
| **Occupation** | Unemployed | 63 | 34 | 2.851 (1.264-6.430) | .012* |
| | Student | 26 | 16 | 2.500 (.981-6.372) | .055 |
| | Housewife | 22 | 24 | 1.410 (.569-3.492) | .457 |
| | Farmer | 52 | 32 | 2.500 (1.095-5.708) | .030* |
| | Self-employee | 52 | 26 | 3.077 (1.326-7.143) | .009* |
| | Government-employee | 13 | 20 | 1 | |
| **Residency** | Urban | 72 | 74 | 1 | |
| | Rural | 156 | 78 | 2.056 (1.346-3.139) | .001* |
| **Level of education** | No formal education | 79 | 40 | 2.962 (1.416-6.198) | .004* |
| | Primary school | 47 | 27 | 2.611 (1.185-5.754) | .017* |
| | Secondary school | 45 | 28 | 2.411 (1.095-5.307) | .029* |
| | Diploma | 41 | 33 | 1.864 (.853-4.070) | .118 |
| | Degree and above | 16 | 24 | 1 | |
| **Leprae reaction** | No | 51 | 66 | 1 | |
| | Type-1 | 98 | 42 | 3.020 (1.806-5.049) | .000* |
| | Type-2(ENL) | 79 | 44 | 2.324 (1.383-3.905) | .001* |
| **No of skin lesion/s** | 0-1 | 48 | 67 | 1 | |
| | 2-3 | 74 | 54 | 1.913 (1.148-3.186) | .013* |
| | 4-5 | 40 | 14 | 3.988 (1.956-8.133) | .000* |
| | 6-7 | 66 | 17 | 5.419 (2.831-10.373) | .000* |
| **Level of skin lesion** | Superficial | 133 | 107 | 1 | |
| | Deep | 88 | 29 | 2.441 (1.494-3.988) | .000* |
| **Mode of symptom onset** | Sudden | 74 | 72 | 1 | |
| | Gradual | 147 | 64 | 2.235 (1.443-3.461) | .000* |
| **Duration of the disease (in years)** | ≤1 | 37 | 57 | 1 | |
| | 2-5 | 93 | 61 | 2.349 (1.390-3.970) | .001* |
| | 6-10 | 76 | 31 | 3.777 (2.098-6.799) | .000* |
| | >10 | 22 | 3 | 11.297 (3.156-40.441) | .000* |
| **WHO classification of leprosy** | PB | 118 | 118 | 1 | |
| | MB | 110 | 34 | 3.235 (2.039-5.133) | .000* |
| **Known history of contact** | No | 165 | 126 | 1 | |
| | Yes | 63 | 26 | 1.850 (1.109-3.089) | .019* |

*Value statistically significant; COR: Crude odds ratio; 1: reference.

The prevalence in the current study was lower than the prevalence of a study done in Ecuador, where the prevalence of leprosy neuropathy was 97.4% [40]. The possible reason for the higher prevalence of Ecuador study might be due to difference in study settings, tool used, and the study design. It was also highly endemic area.

Table 5. Multivariate analysis - Predictors of PN among leprosy patients at BGH 2022, Dessie, Ethiopia.

| Predictorvariables | Category | Occurrence of PN | | Multivariate analysis | |
|---|---|---|---|---|---|
| | | Yes | No | AOR (95% CI) | p-value |
| **Sex** | Male | 137 | 66 | 2.218 (1.230-3.998) | .008* |
| | Female | 91 | 86 | 1 | |
| **Age (in years)** | < 30 | 47 | 81 | 1 | |
| | 30-39 | 48 | 34 | 1.825 (.892-3.733) | >.05 |
| | 40-49 | 66 | 19 | 4.214 (1.973-8.999) | <.001 |
| | ≥ 50 | 67 | 18 | 5.287 (2.473-11.301) | <.001 |
| **Leprosy reaction** | No | 51 | 66 | 1 | |
| | Type-1 | 98 | 42 | 1.982 (1.050- 3.742) | .035* |
| | Type-2(ENL) | 79 | 44 | 2.509 (1.269- 4.962) | .008* |
| **No of skin lesion/s** | 0-1 | 48 | 67 | 1 | |
| | 2-3 | 74 | 54 | 1.563 (.751- 3.254) | >.05 |
| | 4-5 | 40 | 14 | 3.187 (1.115- 9.108) | .031* |
| | 6-7 | 66 | 17 | 3.642 (1.411- 9.403) | .008* |
| **Duration of the disease (in years)** | ≤1 | 37 | 57 | 1 | |
| | 2-5 | 93 | 61 | 1.207 (.577-2.524) | >.05 |
| | 6-10 | 76 | 31 | 1.534 (.651-3.615) | >.05 |
| | >10 | 22 | 3 | 4.866 (1.130-20.958) | .034* |
| **WHO classification of leprosy** | PB | 118 | 118 | 1 | |
| | MB | 110 | 34 | 3.636 (1.431-9.243) | .007* |

*Value statistically significant; AOR: adjusted odds ratio; 1: reference.

On the other hand, the prevalence of the present study was higher than studies conducted in India and Northwest Bangladesh which were 37.9% and 13%, respectively [41,42]. This difference might be explained by the difference in genetic susceptibility, health care qualities and also the study design used, which was prospective cohort in both of those studies.

Inconsistent with the present finding, an institution-based descriptive retrospective study conducted in Colombia on 282 leprosy patients using electronic health records showed that the prevalence of neuropathy was 13.8% [43]. This discrepancy might be because in the present study, most of the patients were from rural areas, low social level, strenuous roads and inaccessibility to health care centers, accounting for this higher prevalence rate.

The prevalence of leprosy peripheral neuropathy reported from Sudan was 42.9%; which was smaller than the current study [44]. The possible reason for this difference might be that the sample size was small, and the study design was prospective cross-sectional for the Sudan study.

According to a study conducted on populations from India, Brazil and Indonesia using the existing individual-based model SIMCOLEP to predict future trends of leprosy incidence that simulates the spread of M. leprae in a population that is structured in households, more than three-quarter of patients had no known history of contact with an index patient [45]. This data is in line with the current study.

However, comparing low-endemic Thailand to high-endemic Bangladesh, Richardus et al. [46] revealed that a smaller percentage of new leprosy patients (one-third) in Thailand had no prior contact history. Ethiopia, where this study was conducted, is a low-endemic country; a higher percentage of patients had no prior contact history.

In the present study, PB leprosy showed high rate than MB cases. But a pattern of predominance of MB forms has been recorded in other similar studies [47–49].

In the present study, 32.4% of patients had type 1 reaction, while 36.8% had a type 2 reaction. Previous studies have found varying prevalence rates of type 1 reactions, ranging from 19% to 30% [50,51].

The percentage of males (53.4%) exceeded that of females (46.6%). In general, leprosy had been more prevalent in males than in females [52]. In different studies from Asian countries such as India and Philippines, the number of male patients had been more than that of the female patients but in some African countries no sex preponderance was observed [53–56]. In the current study, ulnar nerve was the most affected nerve. Similarly, a study done in Colombia showed that ulnar nerve along with the anterior tibial and posterior tibial nerves were mentioned to be the most commonly affected nerves [43]. Croft et al. [57] found that the most commonly affected nerve by function impairment was the posterior tibial (sensory) followed by the ulnar nerve.

Patients seen in the current study were typically diagnosed with leprosy after an extended duration of symptomatic illness, which is consistent with a previous review of Hansen disease patients in north America [58]. The reason for this delay may be related to patients were not seeking care or having limited access to care [59,60].

## Determinants of leprosy peripheral neuropathy

According to the current study, patients age above 40 years appeared to be a predictor of leprosy peripheral neuropathy. This is consistent with national and international studies. For instance, previous study conducted to measure the prevalence of disability and associated factors among registered leprosy patients in all Africa TB and Leprosy Rehabilitation and Training centre (ALERT), Addis Ababa, Ethiopia supported this finding [28]. A study from Colombia also revealed similar result [43].

Studies done in China and Bangladesh revealed an association between old age and leprosy peripheral neuropathy [61,62]. The possible justification for this association might be because of being a chronic complication of leprosy, peripheral neuropathy takes time to develop, so it is expected in older leprotic patients.

In our study, skin lesion was found to be significantly associated with leprosy peripheral neuropathy, where patients with skin lesion were three times at risk to contract leprosy peripheral neuropathy. This finding is supported by a study conducted in northern India in which skin lesion showed 3-4 times risk for leprosy peripheral neuropathy, i.e., the presence of skin lesions was the strongest predictor of subsequent neuropathy [63].

Male sex is found to be more likely associated with leprosy peripheral neuropathy in the present study. Similarly, a study from Indonesia also showed the same association [64]. According to WHO, people with leprosy are predominantly male with which this study is consistent. This might be due to the fact that a greater number of male patients could be attributed to their greater mobility and increased opportunity for contact and are more active in reporting to health facility for seeking treatment. On top of this, male patients may delay care due to long work hours and income instability, which limit opportunities for appointments. The over-representation of males in physical labor jobs could have a dual effect [43]. On the other hand, females cover most of their body parts which could lead to decreased detection of symptoms. These could be the factors responsible for higher chance of male to be a victim of the disease.

MB leprosy was associated with leprosy peripheral neuropathy in current study. This is in line with the study done by Moschioni *et al.* [65], which has shown that multi bacillary leprosy is more prone to neuropathic manifestations. The Bangladesh Acute Nerve Damage Study (BANDS) carried out at the Danish Bangladesh Leprosy Mission also support this finding, where patients with MB leprosy had a 65% risk of developing nerve damage [66].

Older age, delay in diagnosis, thickened nerves at diagnosis, and reversal and ENL reactions were reported as risk factors for the development of leprosy neuropathy by the AMFES study which was done in Ethiopia [67].

In the current study, patients who had leprosy reactions were found to have associated leprosy peripheral neuropathy. This finding is also consistent with previous studies [57,65,67,70]. This finding might be justified by the fact that *M. leprae* affect Schwann cells and the infected cells are susceptible to host immune response and are killed by activated T-cells [68,69]. Besides, loss of nerve function may also be triggered by the mechanical effect of increased pressure within the neural sheath caused by the inflammatory oedema [71].

In this study, it's confirmed that those patients who stayed with leprosy for a longer duration of time are riskier to develop leprosy peripheral neuropathy than does their counterparts. This is consistent with the result of previous study which showed a higher risk of disability in patients who had symptoms for prolonged duration [28]. This could be described by the fact that most of the patients sought medical care very late after the diseases had progressed.

## Limitations of the study

- The study is confined to a limited geographical setting and was conducted over a short period, which may restrict the generalization of our findings to broader populations of leprosy patients

- Inferring casual association is difficult due to the cross-sectional nature of the study and lack of nerve conduction testing, which is the gold standard diagnostic test for confirmation of leprosy peripheral neuropathy diagnosis

- Due to secondary nature of the data, we missed grade of disability at diagnosis, specific details about eye involvement and number of patients with nerve thickening without any sensory, motor or autonomic involvement

## Conclusion

The study explored the prevalence, patterns and determinants of peripheral neuropathy among registered leprosy patients. The prevalence of leprosy peripheral neuropathy was high, 60%. Sensory impairment was the most common pattern of peripheral neuropathy. Male sex, advanced age, presence of leprosy reactions, presence of four and more skin lesions, longer duration of the disease, and MB leprosy were the risk factors associated with leprosy peripheral neuropathy. Prolonged duration of illness is revealed as one of the risk factors for nerve damage, calling for urgent assessment.

## Supporting information

**S1 Data. This is the SPSS data exported to excel.** Analysis was conducted from this data. It contains the values from which the means, standard deviations and other measures are taken and the values used to build graphs and figures.
(XLSX)

## Author contributions

**Conceptualization:** Endris Seid Muhaba, Seid Mohammed Abdu.

**Data curation:** Endris Seid Muhaba.

**Formal analysis:** Endris Seid Muhaba.

**Investigation:** Endris Seid Muhaba.

**Methodology:** Endris Seid Muhaba, Seid Mohammed Abdu.

**Project administration:** Endris Seid Muhaba, Seid Mohammed Abdu.

**Resources:** Endris Seid Muhaba.

**Software:** Endris Seid Muhaba.

**Supervision:** Soressa Abebe Geneti, Dereje Melka.

**Validation:** Soressa Abebe Geneti, Dereje Melka.

**Visualization:** Endris Seid Muhaba.

**Writing – original draft:** Endris Seid Muhaba.

**Writing – review & editing:** Endris Seid Muhaba, Soressa Abebe Geneti, Dereje Melka.

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
