## [Decision Letter · Decision Letter 0]

27 May 2024

Dear Mr. SEID,

Thank you very much for submitting your manuscript "Prevalence, patterns and determinants of peripheral neuropathy among leprosy patients at Boru Meda General Hospital, Dessie, Ethiopia: A retrospective study" for consideration at PLOS Neglected Tropical Diseases. As with all papers reviewed by the journal, your manuscript was reviewed by members of the editorial board and by several independent reviewers. In light of the reviews (below this email), we would like to invite the resubmission of a significantly-revised version that takes into account the reviewers' comments. 

Major revision according to the reviewers' comments is required. Please address every point.

We cannot make any decision about publication until we have seen the revised manuscript and your response to the reviewers' comments. Your revised manuscript is also likely to be sent to reviewers for further evaluation.

Sincerely,

Katja Fischer

Academic Editor

Stuart Blacksell

Section Editor

Major revision according to the reviewers' comments is required. Please address every point.

Reviewer's Responses to Questions

**Key Review Criteria Required for Acceptance?**

**Methods**

-Are the objectives of the study clearly articulated with a clear testable hypothesis stated?

-Is the study design appropriate to address the stated objectives?

-Is the population clearly described and appropriate for the hypothesis being tested?

-Is the sample size sufficient to ensure adequate power to address the hypothesis being tested?

-Were correct statistical analysis used to support conclusions?

-Are there concerns about ethical or regulatory requirements being met?

Reviewer #1: -Are the objectives of the study clearly articulated with a clear testable hypothesis stated?

No, please see my detailed comments.

-Is the study design appropriate to address the stated objectives?

Partially, please see my detailed comments.

-Is the population clearly described and appropriate for the hypothesis being tested?

Partially, please see my detailed comments.

-Is the sample size sufficient to ensure adequate power to address the hypothesis being tested?

Yes

-Were correct statistical analysis used to support conclusions?

I think it is ok.

-Are there concerns about ethical or regulatory requirements being met?

Yes.

Reviewer #2: It is not clear whether diagnosis of leprosy patients was done by a trained person. In absence of it, the data can be misleading.

**Results**

-Does the analysis presented match the analysis plan?

-Are the results clearly and completely presented?

-Are the figures (Tables, Images) of sufficient quality for clarity?

Reviewer #1: -Does the analysis presented match the analysis plan?

I think it its ok. 

-Are the results clearly and completely presented?

Partially, I missed the grade of disability at diagnosis. 

-Are the figures (Tables, Images) of sufficient quality for clarity?

Some data are duplicated in the text and table and figures. It is not necessary.

Reviewer #2: 1. It is not clear whether these patients are newly diagnosed leprosy patients or patients already on Multi-Drug Therapy.

2. In Table 2, all variables are given except disability status at the time of diagnosis, grading of sensory and motor impairment, and whether skin smear was done in a proportion of cases. If this was/is primarily a leprosy centre, then, it should have facilities for skin smear and nerve function assessment.

3. If this data is taken at the time of diagnosis and if all patients are new patients (never treated for leprosy before), then 69.2% of patients presenting with any type of lepra reaction (type 1 or 2) at the time of diagnosis is a very high proportion and requires cross-checking. In the method, two persons have checked the records independently but it is not clear who made the diagnosis (a medical doctor or a nurse) and whether they were properly trained.

4. As per WHO classification, single or more than one nerve involvement due to leprosy is classified as MB. From Tables 2 and 3, it is not clear whether these patients were included in the MB classification.

5. Out of 380 patients, 115 presented with 0-1 skin lesion, and, 263 patients presented with lepra reaction. From this data, it looks, like all patients with more than 1 skin lesion presented with lepra reaction or several of single skin lesion presented with lepra reaction ? this phenomenon is not common.

6. Figure 1, the graphical presentation of the pattern of peripheral neuropathy is not clear. It should be presented in a tabular form. As per table 2, 140 patients (36.8%) presented with type 1 lepra reaction. It is an acute neuritis form but from the graph (figure 1) it shows less than 5% of patients in both sexes with neuropathic pain. Also, approximately, 20% of patients (both males and females) are shown as presenting with a motor impairment which is a WHO grade 2 disability. In the same figure 1, 60% of males and 80% of females had no impairment whereas in Table 3 it is shown that more 58% patients presented with a single or multiple nerve involvement. Also, some patients are shown as a motor impairment solely. How is it possible that in leprosy patients presenting with a motor deficit without sensory involvement. Was a motor neuron disease ruled out as differential diagnosis ? 

7. In several places, leprosy perihpheral nerve involvement vis-à-vis leprous peripheral neuropathy is used. It is better that authors use leprosy beacuse leprous gives an impression of peripheral neuropathy due to lepromatous leprosy. 

8. Text needs to be adjusted and ancient history e.g., leprosy is 4000 years old etc, is not warranted in the manuscript. Also, authors should quote the latest epidemiological figures for global and Ethiopian leprosy status from the annual leprosy surveillance updates from the WHO weekly epidemiological record.

**Conclusions**

-Are the conclusions supported by the data presented?

-Are the limitations of analysis clearly described?

-Do the authors discuss how these data can be helpful to advance our understanding of the topic under study?

-Is public health relevance addressed?

Reviewer #1: -Are the conclusions supported by the data presented?

Partially 

-Are the limitations of analysis clearly described?

No

-Do the authors discuss how these data can be helpful to advance our understanding of the topic under study?

No

-Is public health relevance addressed?

Yes. Peripheral neuropathy caused by leprosy is a public health problem. The present study highlighted the importance of not forgetting leprosy in Ethiopia.

Reviewer #2: presented above. Major revision requested.

**Editorial and Data Presentation Modifications?**

Reviewer #1: (No Response)

Reviewer #2: (No Response)

**Summary and General Comments**

Reviewer #1: Title: Prevalence, patterns, and determinants of peripheral neuropathy among leprosy patients at Boru Meda General Hospital, Dessie, Ethiopia: A retrospective study

Authors: Endris Seid, Soressa Abebe Geneti, Dereje Melka, Seid Mohammed.

Manuscript Number: PNTD-D-24-00127

General comments: This manuscript reports findings of a retrospective study to identify the prevalence, patterns, and determinants of peripheral neuropathy among patients diagnosed with leprosy and seen at the dermatology outpatient clinic of the Boru Meda General Hospital, located in Dessie town, Ethiopia. According to the study findings, the

prevalence of leprous peripheral neuropathy was 60%, and, in general, it was associated with males, age greater than 40 years, delayed diagnosis, leprae reactions, and multibacillary leprosy. 

This study is worth acknowledgment once the work brings more information about leprosy in rural Ethiopia. However, the authors should clarify many aspects before the manuscript can be considered for publication by the PLOS Neglected Tropical Diseases. 

Specific comments:

1. The manuscript needs a revision for English language.

2. The name of the microorganism should be presented in italics, following the writing standard for scientific names of biological species.

3. The introduction is too generic; it needs to be more specific and guide the reader towards the main objective of the study. The introduction shouldn't only present well-described characteristics about the pathophysiology and classification of leprosy. I suggest it be rewritten to justify the importance and relevance of the study. For example: Ethiopia has declared the elimination of leprosy as a public health problem, yet new cases continue to be diagnosed annually. How many cases are in children? What percentage of grade 2 physical disability at diagnosis? What are the weaknesses of the leprosy control program in Ethiopia and how could this perpetuate the disease transmission in the country? These questions need to be addressed, either in the introduction or in the discussion.

4. I think it's important to describe the study setting, the characteristics of the region, and any access and communication difficulties. Could any cultural aspect explain the results found?

5. The literature review needs to be updated. I suggest using the most recent epidemiological data published by the WHO.

6. North America does not only have leprosy in immigrants. Please, see:

Belzer A, Ochoa MT, Adler BL. Autochthonous Leprosy in the United States. N Engl J Med. 2023 Jun 29;388(26):2485-2487. doi: 10.1056/NEJMc2302317. PMID: 37379141.

7. Majority is a subset of a group whose number is greater than half of the entire group. In the manuscript it is used with a different meaning.

8. The authors stated that “more than three-quarters of patients, 291(76.6%) had no contact history with leprosy inhabitants previously”. It's important to highlight that they had no known contact with any person affected by leprosy.

9. Table 3 repeats numbers already presented in the text. This is unnecessary.

10. Out of 228 affected nerves. These are not affected nerves, but affected individuals.

11. Leprosy is primarily a neural disease; all patients present some degree of clinical or subclinical neuropathy. This needs to be made clear in the text.

12. There is no description of the degree of physical disability. This would be a more concrete data to present.

13. The authors need to inform how the active case-finding system works in the study area. Without contact, there is no transmission of M. leprae.

14. The discussion presents some hypotheses apparently without scientific basis. For example: “The predominance of PB forms in our population may be due to relatively good immunity against leprosy.” I think the most plausible hypothesis in this case is that the patients were misclassified as PB.

15. The authors should cite this important systematic review and meta-analysis about the risk factors for physical disability in patients with leprosy. 

de Paula HL, de Souza CDF, Silva SR, et al. Risk Factors for Physical Disability in Patients With Leprosy: A Systematic Review and Meta-analysis. JAMA Dermatol. 2019;155(10):1120–1128. doi:10.1001/jamadermatol.2019.1768

16. Replace references 6 and 7 with Weekly epidemiological record 15 SEPTEMBER 2023, 98th YEAR / No 37, 2023, 98, 409–430 

chrome-extension://efaidnbmnnnibpcajpcglclefindmkaj/ https://cdn.who.int/media/docs/default-source/weekly-epidemiological-record/wer9837-eng-fre.pdf

17. Reference 61 = 58, it's repeated.

Reviewer #2: (No Response)

PLOS authors have the option to publish the peer review history of their article (what does this mean? ). If published, this will include your full peer review and any attached files.

**Do you want your identity to be public for this peer review?** For information about this choice, including consent withdrawal, please see our Privacy Policy .

Reviewer #1: No

Reviewer #2: Yes: Saurabh Jain
---

## [Decision Letter · Decision Letter 1]

21 Sep 2024

Dear Mr. SEID,

Thank you very much for submitting your manuscript "Prevalence, Prevalence, patterns and determinants of peripheral neuropathy among leprosy patients: A retrospective study" for consideration at PLOS Neglected Tropical Diseases. As with all papers reviewed by the journal, your manuscript was reviewed by members of the editorial board and by several independent reviewers. In light of the reviews (below this email), we would like to invite the resubmission of a significantly-revised version that takes into account the reviewers' comments. 

Please address the comments of the reviews. While you have done this in the first round of revisions, it would appear to be insufficient. Please ensure that your revised manuscript addresses all of the reviewers concerns.

We cannot make any decision about publication until we have seen the revised manuscript and your response to the reviewers' comments. Your revised manuscript is also likely to be sent to reviewers for further evaluation.

Sincerely,

Stuart D. Blacksell

Section Editor

Stuart Blacksell

Section Editor

Please address the comments of the reviews. While you have done this in the first round of revisions, it would appear to be insufficient. please ensure that your revised manuscript addresses all of the reviewers concerns.

Reviewer's Responses to Questions

**Key Review Criteria Required for Acceptance?**

**Methods**

-Are the objectives of the study clearly articulated with a clear testable hypothesis stated?

-Is the study design appropriate to address the stated objectives?

-Is the population clearly described and appropriate for the hypothesis being tested?

-Is the sample size sufficient to ensure adequate power to address the hypothesis being tested?

-Were correct statistical analysis used to support conclusions?

-Are there concerns about ethical or regulatory requirements being met?

Reviewer #1: -Are the objectives of the study clearly articulated with a clear testable hypothesis stated?

Yes.

-Is the study design appropriate to address the stated objectives?

Yes.

-Is the population clearly described and appropriate for the hypothesis being tested?

Yes.

-Is the sample size sufficient to ensure adequate power to address the hypothesis being tested?

Yes,

-Were correct statistical analysis used to support conclusions?

I think it is ok.

-Are there concerns about ethical or regulatory requirements being met?

I think it is ok.

Reviewer #2: -Authors need to adjust the text according to the patients (new) presenting with peripheral neuropathy or with lepra reaction at the time of diagnosis and those who worsened during the course of MDT or after completion of treatment. 

-Authors should also provide a separate table on peripheral neuropathy, lepra reaction vis-a-vis the type of disease (PB, MB).

-Nerve involvement in table 2 (clinical characteristics of patients) is missing. In the same table duration of disease should be qualified with at the time of presentation as a new case. 

- Please specify when you say that this hospital sees 20 leprosy patients per month. Does this mean 20 new cases or it includes cured patients coming back for their nerve damage consultation?

- It is not clear why authors avoided to follow WHO grading of voluntary muscle testing. 

- Authors should add a separate table on sensory and voluntary muscle testing grades for eyes, hands and feet. It is surprising that none of the patient presented with eye involvement. 

- Since authors have not followed the latest WHO criteria to classify patients as PB and MB, they should mention it in the methods or as a separate note on limitations of the study. 

- Whether nerve thickening without any sensory, motor or autonomic involvement is taken into consideration.

- How many patients were of pure neuritic leprosy (no skin involvement)?

**Results**

-Does the analysis presented match the analysis plan?

-Are the results clearly and completely presented?

-Are the figures (Tables, Images) of sufficient quality for clarity?

Reviewer #1: -Does the analysis presented match the analysis plan?

I think it its ok. 

-Are the results clearly and completely presented?

Partially, I missed the grade of disability at diagnosis. However, it seems the authors do not have this data. So, once they declare it as a study limitation in the discussion, it will be ok for me.

-Are the figures (Tables, Images) of sufficient quality for clarity?

Ok.

Reviewer #2: --Authors should also provide a separate table on peripheral neuropathy, lepra reaction vis-a-vis the type of disease (PB, MB).

-On page 8, out of 228 affected individuals, 94 (41.2%) patients' nerve impairments are described as not specified. What does this mean ? Either patient is suffering from nerve involvement or not suffering. If this is not specified, then why this number has been taken into calculations.

**Conclusions**

-Are the conclusions supported by the data presented?

-Are the limitations of analysis clearly described?

-Do the authors discuss how these data can be helpful to advance our understanding of the topic under study?

-Is public health relevance addressed?

Reviewer #1: -Are the conclusions supported by the data presented?

Yes.

-Are the limitations of analysis clearly described?

Can be improved. Please see my detailed comments

-Do the authors discuss how these data can be helpful to advance our understanding of the topic under study?

Ok.

-Is public health relevance addressed?

Yes. Peripheral neuropathy caused by leprosy is a public health problem. The present study highlighted the importance of not forgetting leprosy in Ethiopia.

Reviewer #2: - In the abstract, authors should mention the prevalence finding of this analysis.

- It is not clear why authors are mentioning 10-15 years old leprosy data of the World and Ethiopia and why not giving the latest figures that is readily available publicly in WHO's weekly epidemiological report on global leprosy surveillance. Authors seem to be aware about availability of this data as they have already given this reference but from previous years.

- In the Introduction section, WHO figures have been given without the reference. 

- WHO target of leprosy elimination of less than 1 case per 10,000 population is defined as a public health problem target and not as a elimination of disease (zero case). Therefore, new cases are expected to come for foreseeable future.

- On page 2, last paragraph, please include lateral popliteal nerve as one of the affected nerves.

- If more than 40 years and above is one of the determinants then how authors have ruled out peripheral neuropathy due to micronutrient deficiencies (both in males and females) and whether history of smoking and tobacco is taken?

- Figure 1 distribution of pattern of leprosy peripheral neuropathy: In leprosy, sensory impairment is the first sensation to go, therefore, motor impairment without sensory impairment should be ruled out for a motor neuron disease. Section on patterns of neuropathy needs more clarification and description. From this graph, and from the tables, it is not clear whether steroids were prescribed in lepra reaction patients and what was the impact of it on the nerve impairments. Generally, with an adequate course, majority patients improve without any residual disabilities specially acute neuritis cases. It is also important to note, how type 2 lepra reaction patients were managed? whether clofazimine were prescribed to them and what was the outcome. 

- Proof reading is needed for better clarity on drafting.

- Overall major improvements are needed in the manuscript. 

- Several of references are quoted from previous articles from this centre. This should be avoided and original references should be given.

**Editorial and Data Presentation Modifications?**

Reviewer #1: Specific comments:

1. Change “leprae reaction” to “leprosy reaction”. 

2. Leprosy is primarily a neural disease; all patients present some degree of clinical or subclinical neuropathy. This needs to be made clear in the text. In the outcomes definitions, leprosy peripheral neuropathy is defined as: “functional impairment and/or structural damage to sensory, motor and autonomic nerve fibres within the peripheral nervous system.”

I suggest the authors should include the following: “detected by clinical nerve palpation (enlarged nerve, pain or feeling of electric shock)” without the use of ultrasound, electroneuromyography, or any complementary examination. 

3. There is no description of the degree of physical disability. This would be a more concrete data to present.

I think it should be mandatory to talk about physical disabilities when the authors intend to investigate the prevalence and patterns of leprosy neuropathy. Once it seems the authors do not have these data, I suggest including this in the discussion as a study limitation.

Reviewer #2: A major reorientation in the data presentation is needed as per the title of the manuscript.

**Summary and General Comments**

Reviewer #1: Specific comments:

1. Change “leprae reaction” to “leprosy reaction”. 

2. Leprosy is primarily a neural disease; all patients present some degree of clinical or subclinical neuropathy. This needs to be made clear in the text. In the outcomes definitions, leprosy peripheral neuropathy is defined as: “functional impairment and/or structural damage to sensory, motor and autonomic nerve fibres within the peripheral nervous system.”

I suggest the authors should include the following: “detected by clinical nerve palpation (enlarged nerve, pain or feeling of electric shock)” without the use of ultrasound, electroneuromyography, or any complementary examination. 

3. There is no description of the degree of physical disability. This would be a more concrete data to present.

I think it should be mandatory to talk about physical disabilities when the authors intend to investigate the prevalence and patterns of leprosy neuropathy. Once it seems the authors do not have these data, I suggest including this in the discussion as a study limitation.

Reviewer #2: This work and the observations are important for leprosy control in Ethiopia but data should be presented in line with the title of the article.

PLOS authors have the option to publish the peer review history of their article (what does this mean? ). If published, this will include your full peer review and any attached files.

**Do you want your identity to be public for this peer review?** For information about this choice, including consent withdrawal, please see our Privacy Policy .

Reviewer #1: Yes: Josafá Gonçalves Barreto

Reviewer #2: Yes: Saurabh Jain
---

## [Decision Letter · Decision Letter 2]

17 Dec 2024

PNTD-D-24-00127R2Prevalence, patterns and determinants of peripheral neuropathy among leprosy patients: A retrospective studyPLOS Neglected Tropical Diseases Dear Dr. SEID, Thank you for submitting your manuscript to PLOS Neglected Tropical Diseases. After careful consideration, we feel that it has merit but does not fully meet PLOS Neglected Tropical Diseases's publication criteria as it currently stands. Therefore, we invite you to submit a revised version of the manuscript that addresses the points raised during the review process. Please submit your revised manuscript within 30 days Jan 16 2025 11:59PM. If you will need more time than this to complete your revisions, please reply to this message or contact the journal office at plosntds@plos.org. Please include the following items when submitting your revised manuscript:* A rebuttal letter that responds to each point raised by the editor and reviewer(s). You should upload this letter as a separate file labeled 'Response to Reviewers '. This file does not need to include responses to any formatting updates and technical items listed in the 'Journal Requirements' section below.* A marked-up copy of your manuscript that highlights changes made to the original version. You should upload this as a separate file labeled 'Revised Manuscript with Track Changes '.* An unmarked version of your revised paper without tracked changes. You should upload this as a separate file labeled 'Manuscript '. If you would like to make changes to your financial disclosure, competing interests statement, or data availability statement, please make these updates within the submission form at the time of resubmission. Guidelines for resubmitting your figure files are available below the reviewer comments at the end of this letter. We look forward to receiving your revised manuscript. Kind regards, Stuart D. BlacksellSection EditorPLOS Neglected Tropical Diseases

Shaden Kamhawi

co-Editor-in-Chief

Paul Brindley

co-Editor-in-Chief

 **Additional Editor Comments :** Please see the comments from Reviewer #3 below for the Results and the Conclusions sections. Please address all of these issues completely and please indicate how these have been addressed in the response to the reviewers.  **Journal Requirements:**

1) Please upload the main figure as a separate Figure file in .tif or .eps format. For more information about how to convert and format your figure files please see our guidelines: 

2) Please ensure that all Table files have corresponding citations and legends within the manuscript. Currently, table 3 in your submission file inventory does not have an in-text citation. Please include the in-text citation of the table.

3) Tables should not be uploaded as individual files. Please remove these files and include the Tables in your manuscript file as editable, cell-based objects. For more information about how to format tables, see our guidelines:

https://journals.plos.org/plosntds/s/tables 

4) We note that your Data Availability Statement is currently as follows: "All data underlying their findings are fully available without restriction". Please confirm at this time whether or not your submission contains all raw data required to replicate the results of your study. Authors must share the “minimal data set” for their submission. PLOS defines the minimal data set to consist of the data required to replicate all study findings reported in the article, as well as related metadata and methods (https://journals.plos.org/plosone/s/data-availability#loc-minimal-data-set-definition).

**Reviewers' comments:**

**Results**

Reviewer #3: -Does the analysis presented match the analysis plan? Yes

-Are the results clearly and completely presented? some confusion is there eg. skin lesions 0-1 is labelled in Table 2, but this does not specify how many had pure neuritic form of leprosy. It should have no skin lesion to make diagnosis of pure neuritic form of leprosy.

-Are the figures (Tables, Images) of sufficient quality for clarity? same as the above answer- some confusions are there.

**Conclusions**

Reviewer #3: -Are the conclusions supported by the data presented? Yes

-Are the limitations of analysis clearly described? Yes

-Do the authors discuss how these data can be helpful to advance our understanding of the topic under study? Yes

-Is public health relevance addressed? Yes

**Editorial and Data Presentation Modifications?**

Reviewer #3: Typing error needs to be corrected. 

**Figure resubmission:**

While revising your submission, please upload your figure files to the Preflight Analysis and Conversion Engine (PACE) digital diagnostic tool, https://pacev2.apexcovantage.com/. PACE helps ensure that figures meet PLOS requirements. To use PACE, you must first register as a user. Registration is free. Then, login and navigate to the UPLOAD tab, where you will find detailed instructions on how to use the tool. If you encounter any issues or have any questions when using PACE, please email PLOS at figures@plos.org. Please note that Supporting Information files do not need this step. If there are other versions of figure files still present in your submission file inventory at resubmission, please replace them with the PACE-processed versions. **Reproducibility:** To enhance the reproducibility of your results, we recommend that authors of applicable studies deposit laboratory protocols in protocols.io, where a protocol can be assigned its own identifier (DOI) such that it can be cited independently in the future. Additionally, PLOS ONE offers an option to publish peer-reviewed clinical study protocols. Read more information on sharing protocols at https://plos.org/protocols?utm_medium=editorial-email&utm_source=authorletters&utm_campaign=protocols

---

## [Editor Report · Decision Letter 3]

25 Feb 2025

Dear Mr. SEID,

We are pleased to inform you that your manuscript 'Prevalence, patterns and determinants of peripheral neuropathy among leprosy patients: A retrospective study' has been provisionally accepted for publication in PLOS Neglected Tropical Diseases.

Best regards,

Stuart D. Blacksell

Section Editor

Shaden Kamhawi

co-Editor-in-Chief

Paul Brindley

co-Editor-in-Chief

---

## [Editor Report · Acceptance letter]

Dear Mr. SEID,

We are delighted to inform you that your manuscript, "Prevalence, patterns and determinants of peripheral neuropathy among leprosy patients: A retrospective study," has been formally accepted for publication in PLOS Neglected Tropical Diseases.

Best regards,

Shaden Kamhawi

co-Editor-in-Chief

Paul Brindley

co-Editor-in-Chief
